# Vestibular Disorders after Kidney Transplantation: Focus on the Pathophysiological Mechanisms Underlying the Vertical Nystagmus Associated with Tacrolimus-Related Hypomagnesamia

**DOI:** 10.3390/ijerph19042260

**Published:** 2022-02-16

**Authors:** Pasquale Viola, Vincenzo Marcelli, Domenico Sculco, Davide Pisani, Alfredo Caglioti, Filippo Ricciardiello, Alfonso Scarpa, Alessia Astorina, Giuseppe Tortoriello, Luca Gallelli, Giovambattista De Sarro, Giuseppe Chiarella

**Affiliations:** 1Unit of Audiology, Department of Experimental and Clinical Medicine, Regional Centre of Cochlear Implants and ENT Diseases, Magna Graecia University, 88100 Catanzaro, Italy; pasqualeviola@unicz.it (P.V.); alessiaastorina7@gmail.com (A.A.); chiarella@unicz.it (G.C.); 2UOC ENT Ospedale del Mare, 80147 Naples, Italy; vincenzo.marcelli@hotmail.it (V.M.); dott.giuseppetortoriello@virgilio.it (G.T.); 3Unit of Audiology, Mater Domini University Hospital, 88100 Catanzaro, Italy; d.sculco.mico@alice.it; 4Nephrology and Dialysis Unit, Mater Domini University Hospital, 88100 Catanzaro, Italy; alfredocaglioti@libero.it; 5ENT Department, AORN Cardarelli, 80131 Napoli, Italy; filipporicciardiello@virgilio.it; 6Department of Medicine and Surgery, University of Salerno, 84084 Fisciano, Italy; alfonsoscarpa@yahoo.it; 7Operative Unit of Pharmacology and Pharmacovigilance, “Mater Domini” Hospital, Department of Health Science, University Magna Graecia, 88100 Catanzaro, Italy; gallelli@unicz.it (L.G.); desarro@unicz.it (G.D.S.); 8Department of Health Science, Research Center FAS@UMG, University of Catanzaro, 88100 Catanzaro, Italy

**Keywords:** vertical nystagmus, hypomagnesemia, tacrolimus, kidney transplantation, vestibular disorder

## Abstract

The purpose of this paper is to present the case of a patient undergoing kidney transplantation who developed limb tremor dizziness and vertical nystagmus (ny) during Tacrolimus (TAC) therapy and to investigate the pathophysiological mechanisms underlying the balance disorder. This case study regards a 51-year old kidney transplant male patient with hand tremors and lower limbs asthenia associated with dizziness and nausea. The symptoms started two months after the beginning of intravenous TAC for renal transplantation. The pure-tone audiometry showed a mild symmetrical high-frequencies down-sloping sensorineural hearing loss. Acoustic emittance measures showed a normal tympanogram; stapedial reflexes were normally elicited. The Auditory Brainstem Responses (ABR) and Cervical Vestibular Evoked Myogenic Potentials (c-VEMPs) were bilaterally normally evoked. The bedside vestibular examination showed spontaneous down-beating stationary persistent, omni-positional nystagmus, not inhibited by fixation. The Head-Shaking Test accentuates the spontaneous ny. The horizontal clinical head impulse test was negative, bilaterally. A biochemical blood test revealed a decrease in Magnesium (Mg) levels (0.8 mg/dL; normal range 1.58–2.55). The integration of Mg induced both a plasma levels normalization and an improvement of clinical symptoms. This case suggests that TAC treatment can induce a Mg depletion that caused the transient cerebellar lesion. Therefore, the monitoring of serum electrolytes during immunosuppressive treatment appears to be a useful tool in order to reduce the central system symptomatology.

## 1. Introduction

Magnesium (Mg) is a cofactor in several enzymatic reactions, also involved in the electrical potential’s maintenance of cell membranes [1]. Hypomagnesemia (serum Mg concentration <1.8 mg/dL; <0.70 mmol/L) causes a wide range of clinical features, predominantly due to cardiological and neurological impairment (e.g., seizures, arrhythmias tremor, aphasia, hemiparesis, ataxia, and dizziness). Several drugs, such as proton pump inhibitors, loop-diuretics, chemotherapy (cisplatin), and immunosuppressant drugs (cyclosporin and TAC), can cause mild or severe hypomagnesaemia [2,3]. We report a patient undergoing kidney transplantation with the onset of limb tremor and dizziness during TAC treatment, and we also investigate the pathophysiological mechanism underlying the balance disorders linked to Mg decrease.

## 2. Case Report

A 51-year-old male patient with a diagnosis of metabolic syndrome (obesity, high blood pressure, diabetes mellitus, and dyslipidemia), chronic renal failure (2009), acute pancreatitis (2011), cholecystectomy (2013), and kidney transplant (2019) was sent to our unit (June 2020) for hand tremors and lower limbs asthenia, associated with dizziness and nausea. The anamnesis revealed that after kidney transplant, the patient received, in agreement with the literature [4], a first induction with 100 mg/day of intravenous antithymocyte globulin (ATG) for three days and then IV therapy with Prednisone (20 mg/day for 3 weeks, with a 5 mg tapering every 2 weeks), Mycophenolate Mofetil (1 gr/day) and TAC (23 mg/day) in order to maintain blood concentration in the normal range (10–12 ng/mL).

After transplant, the patient gained excellent immediate functional results (serum creatinine 1.5–1.8 mg/dL), but in April 2020, he developed an impairment of kidney function. Renal biopsy diagnosed an acute severe cellular interstitial rejection, mild acute vascular rejection evolved into active chronic cellular rejection, and chronic transplant artery disease. Therefore, a treatment with ATG (30 mg/day for 5 days) plus prednisone (20 mg/day IV for 3 weeks, tapered by 5 mg every 2 weeks) was started.

Two months later (June 2020), the patient came to our unit with hand tremors and lower limbs asthenia with dizziness and nausea, therefore an extensive audio-vestibular examination was conducted.

Air and bone conduction Pure Tone Audiometry (frequencies 125–8000 Hz using Piano Clinical Audiometer; Inventis, Padua, Italy) showed a mild symmetrical high-frequencies down-sloping sensorineural hearing loss. Acoustic emittance measures, performed to evaluate the functional integrity of the eardrum and middle ear, using tympanometry and acoustic stapedial reflex tests (AT 235 Tympanometer Interacoustics, Denmark), showed a normal tympanogram (type A) with stapedial reflexes normally elicited.

The ABR procedure, performed at 0.1 ms click stimulus and alternated polarity (21 pps rate with 2000 repetitions, acquisition interval of 10 ms with HP of 100 Hz and LP of 3000 Hz; ICS Chartr EP 200, Otometrics, Taastrup, Denmark) were bilaterally normally evoked as well as the c-VEMPs. These were performed using 500 Hz logon with rarefaction polarity, delivered through an insert earphone starting at 130 dB SPL and reaching thresholds. An average of 100 responses was recorded for each run, in supine position and the head rotated sideways towards one shoulder to activate the sternocleidomastoid muscle. The c-VEMPs responses were measured by monaural acoustic stimulation with ipsilateral recording (ICS Chartr EP 200, Otometrics, Taastrup, Denmark).

The bedside vestibular examination performed by video-goggles (GN Otometrics, Taastrup, Denmark) showed a spontaneous down-beating stationary persistent omni-positional ny, not inhibited by fixation. The Head-Shaking Test (HST) accentuated the spontaneous ny, and the horizontal clinical head impulse test (c-HIT) was negative bilaterally.

The Video Head Impulse Test (v-HIT) performed using ICS-impulse^®^ equipment (GN Otometrics, 99 Taastrup, Denmark) for the lateral semicircular canals (LSC) showed increased vestibulo-oculomotor reflex (VOR) gain for the right LSC only (1.39), while the VOR gain for the left LSC was normal (0.94) (Figure 1).

The impact of vertigo on quality of life (QoL) was assessed through the Dizziness Handicap Inventory (DHI) [5] that documented a severe handicap (score of 72 points).

Biochemical evaluation of a blood sample documented a significant decreased level of Mg (0.8 mg/dL; range 1.58–2.55); therefore, a Mg integration (5 mg/day, os) was added with a rapid improvement of dizzy symptoms, hand tremors, and asthenia.

In October 2020, both the clinical and biochemical evaluation documented the absence of clinical symptoms and normal range of blood Mg (1.6 mg/dL) even if PTA, acoustic emittance, ABR, and c-VEMPs were unchanged. The bedside vestibular examination showed absence of spontaneous ny, position, and positioning ny. The HST and horizontal clinical HIT were negative. The v-HIT showed normal VOR gain for the right (1.09) and left (0.98) LSC (Figure 2). The DHI indicated a mild handicap (score 18 points). This clinical picture was compatible with a deficit of the vestibulo-cerebellar system probably related to hypomagnesaemia.

## 3. Discussion

We reported the case of a man who developed clinical neurological symptoms probably related to adecreased plasma Mg levels during the treatment with TAC for kidney transplant. [6]. Hypomagnesemia occurs frequently after kidney transplantation during immunosuppressive treatment with Calcineurin inhibitors (CNI). Even if the mechanisms leading to hypomagnesemia are not fully understood, CNI could cause a decrease in the transcriptional expression of the Mg transporter in the distal collecting tubule leading to a reduced reabsorption of Mg [7].

Mg is mainly absorbed through the small intestine. The Mg transport systems are a passive paracellular mechanism and transcellular transport by dedicated Mg channels and transporters. In particular, member 1 of solute carrier family 41, magnesium transporter 1, and transient receptor potential melastatin type 6 and 7 have been described.

Mg homeostasis is maintained, under hormonal control, by the intestine, bone, and kidney. Serum Mg is filtered by the renal glomeruli and then reabsorbed along the nephron. Magnesium transport across cell membranes shows tissue variability and, among the body’s tissues, is higher in the heart, liver, kidney, skeletal muscle, red cells, and brain. Thus, magnesium transport, the physiology of magnesium homeostasis, and metabolic activity of the cell are strictly correlated.

CNI such as TAC or Cyclosporine cause a renal loss of Mg through the downregulation of the transient receptor potential melastatin subtype 6, which is an ion channel present on the epithelial cell membranes of the gastrointestinal and renal systems and which allows the active reabsorption of Mg.

Hypomagnesemia is commonly reported during TAC treatment [8]. However, it has to be noted that TAC administration induces lower nephrotoxic, neurologic, and cardiovascular adverse drug reactions than cyclosporine.

Hypomagnesemia may be present together with nonspecific symptoms such as anorexia and nausea, cardiological abnormalities such as arrhythmias, and with neurological symptoms and signs [9,10]. A further, often under-recognized, clinical manifestation of Mg depletion is vertical, usually down-beating, ny [11,12,13].

To fully understand our clinical case, it is necessary to describe the possible pathophysiological mechanisms underlying a down-beating nystagmus (DBN).

First, we need to consider the intrinsic asymmetry on the sagittal plane that characterizes both peripheral and central vestibular apparatus.

In particular, the anatomical orientation of the semicircular canals (SCs) and the physiological vestibule-oculomotor responses in the sagittal plane, determine the genesis of a slow phase directed upwards that must be constantly controlled in an inhibitory sense by the cerebellar structures. On the other hand, the Purkinje cells (PCs) of the flocculus also show a functional asymmetry, which promotes the genesis of a slow phase, directed downwards. In summary, the asymmetry on the vertical plane is characterized by mechanisms that favor a slow phase directed upwards and by mechanisms that favor a slow phase directed downwards. The lack of cerebellar control of these mechanisms will lead to a DBN, whose causes may be the following:The involvement of the VOR in the vertical plane, secondary to a cerebellar dysfunction. It probably represents the most common cause of vestibular down-beat syndrome, and it is possible to recognize pathogenetic mechanisms involving PCs, the Y group of vestibular nuclei, and the flocculus PCs with down direction.Lesion of the cerebellum-brainstem circuit of the neural integrator for the control of the eccentric gaze in the vertical plane, with involvement of the neural integrator. This is a complex circuit that includes the nucleus prepositus hypoglossi, the medial vestibular nucleus (the so-called NPH-MVN area), the interstitial nucleus of Cajal and the vestibule-cerebellum, where the efferences of the paramedian tract (PMT) cells arrive. A lesion of this circuit and the consequent increase in the time constant would cause the eyes to slide upwards in an eccentric position: the appearance of rapid reset phases will lead to the DBN [14,15,16,17].Alteration of cerebellum-maculo-canalar reflexes. The imbalance of maculo-canalar pathways would justify the modulation of the ny by the gravitational vector [18,19,20]. In this regard, the control that the nodulus and uvula exert on macular input is well known, a lesion affecting these structures would therefore demodulate their function.Cerebellar lesion of the SmP pathways in the vertical plane. In this regard, we have to consider two possible mechanisms. First, a tonic imbalance of the vertical SmP with asymmetrical alteration seems to be responsible for the DBN. In physiological conditions, even in this case, there is a prevalence of upwards SmP, which is tonically balanced by downwards SmP. A lesion of the cerebellar SmP pathways, vestibular nuclei or prepositus nucleus would result in a reduced downward SmP activity releasing the upward SmP from tonic control and generating a DBN. The role of this asymmetry seems to be highlighted by the fact that normal subjects can develop a DBN after performing an upward SmP for a certain period of time but not after performing a downward SmP [21,22,23]. Another mechanism could be linked to the flocculus: more than 90% of the PCs of the flocculus involved in SmP show a prevalent down activity. A lesion of the flocculus in the presence of such asymmetry due to a prevalent downward activity will cause the eyes to slide upwards, thus generating a DBN [24,25].Extracerebellar lesion. A damage of the fourth ventricle floor may involve the vertical VOR pathways starting from the posterior SC, whose input reaches the medial and lateral vestibular nuclei and the contralateral medial longitudinal fascicle. A lesion of this pathway, involving the vestibular nuclei, will lead to the disappearance of the tonic activity that the posterior SC exerts, first through the vestibular nuclei and the oculomotor nucleus, then on the inferior rectus muscle. Thus, the functional prevalence of the superior rectus muscle will cause the eyes to slide up leading to a DBN [26]. Obviously, the inevitable involvement of the commissural fibers by contiguity, which ensures compensation, explains the chronicity of the syndrome, while the possible involvement of the medial longitudinal fascicle explains any associated internuclear ophthalmoplegia.

Finally, the increased amplitude of the DBN after horizontal HST, also called perverted nystag, as occurred in our patient, is another sign of cerebellar dysfunction. Under physiological conditions, the nodulus-uvula complex (NUC) “couples” the response plane with the stimulus plane. Therefore, a horizontal HST can only generate a ny in the same plane. Otherwise, if the horizontal HST generates a ny on the vertical plane (down- or up-beating ny) or frontal (clockwise or counterclockwise torsional ny), which significantly prevails over any concomitant horizontal ny, or the plane variation of a spontaneous horizontal ny, configures a “perverted” ny. In this case, the NUC dysfunction may provoke the inability to couple the response plane with the stimulus plane [27].

## 4. Conclusions

The hypothesis that hypomagnesemia is a consequence of a vicious circle between the patient’s basic condition of metabolic syndrome and the action of the TAC, on the one hand, and renal dysfunction after transplantation that required additional therapy, on the other, is the most likely. However, the mechanism leading to vestibular disorder is the same. Voltage-dependent Mg blockade of excitatory calcium influx through the NMDA receptor, known to be highly represented in human cerebellar granule cells, has been suggested to be a potential mechanism of hypomagnesemia-induced DBN. Thus, we can hypothesize that the severe Mg depletion experienced by our patient caused the transient cerebellar damage, known to be involved in the genesis of DBN. This transient cerebellar dysfunction could determine the lack of inhibition of only one labyrinthine hemisystem, explaining the evident increase in the right semicircular VOR gain, which normalized after administration of Mg.

## Figures and Tables

**Figure 1 ijerph-19-02260-f001:**
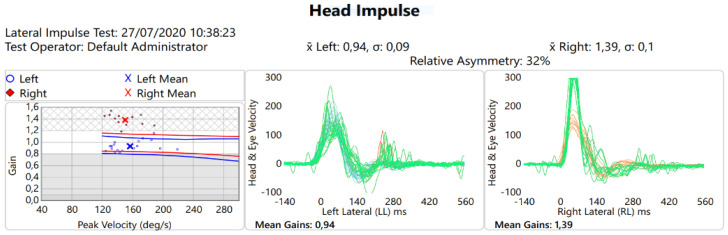
v-HIT: the test shows an asymmetry of VOR gain, with greater values on right side (∆ mean gains: +0.45). In the first graph, the left ear is represented in blue, the right ear in red. The values of the individual measurements are indicated with points, while the average is indicated with an x. The second graph represents the left vestibular activity, the VOR is indicated in green, the saccades are indicated in red, the movement of the head in blue. The third graph represents the right vestibular activity, the VOR is represented in green, the saccades in red and the movements of the head in orange.

**Figure 2 ijerph-19-02260-f002:**
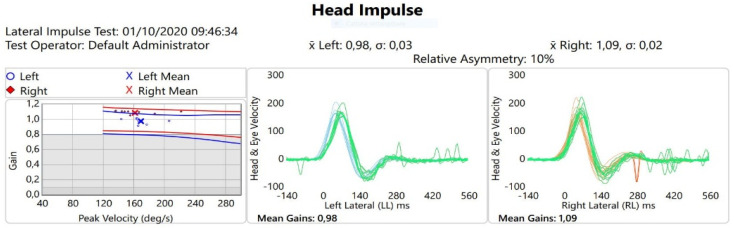
v-HIT: the test documented a reduction in previous gain asymmetry, after Mg oral administration (∆ mean gains: +0.11). In the first graph, the left ear is represented in blue, the right ear in red. The values of the individual measurements are indicated with points, while the average is indicated with an x. The second graph represents the left vestibular activity, the VOR is indicated in green, the saccades are indicated in red, the movement of the head in blue. The third graph represents the right vestibular activity, the VOR is represented in green, the saccades in red and the movements of the head in orange.

## Data Availability

Not applicable.

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
