# Peer review of "Vestibular Disorders after Kidney Transplantation: Focus on the Pathophysiological Mechanisms Underlying the Vertical Nystagmus Associated with Tacrolimus-Related Hypomagnesamia"

_ijerph, 2022, doi:10.3390/ijerph19042260_

Round 1

Reviewer 1 Report

The authors present a case report of a patient who underwent kidney transplantation and received Tacrolimus for graft rejection. The patient developed limb tremors and dizziness two months after Tacrolimus. Extensive auditory/vestibular tests were performed. Abnormal finding was noted only in v-HIT.  Hypomagnesemia was detected in blood test. Upon magnesium supplementation, the patients vestibular symptoms quickly improved. The authors provide a rationale to justify their conclusion that the patients vestibular symptoms were due to cerebellar effects of the hypomagnesemia. Some minor comments:

1) It's mentioned in the abstract that the patient received Tacrolimus. In the main case report (section 2), there is no mention of Tacrolimus.

2) It doesn't look like DBN is defined prior to use in line 134.

3) Unnecessary hyphen for: associated, intravenous, persistent and spontaneous in the abstract.

Reviewer 2 Report

This paper reports the process of diagnosis and treatment of neurological abnormalities and equilibrium dysfunction caused by hypomagnesamia.

The paper will be of interest to the readers of this journal. However, there are a few improvements that should be made before publication.

# Case report; The author's area of expertise is well described. The information of all therapeutic medication will suggest to consider the risk of hypomagnesamia in the case.

# In discussion or introduction;  The author's area of expertise is well discussed. Magnesium metabolism should be additionally described to improve the paper. It will help to understand the mechanism that calcineurin inhibitors (CNI) such as tacrolimus  cause low magnesium.  Are there the problem of gastrointestinal absorption, reabsorption disturbance of renal tubules?  I recommend to add the information about relationship of  CNI and hypomagnesamia especially combined interaction with other drugs.

  It will become more interesting paper to pay attention magnesium for otolaryngologists, neurologists, and doctors using CNI.

Minor comment for line65: Is daily Tacrolimus is 23mg/day not ng/day ?

Round 2

Reviewer 2 Report

Thank you for your revision. The manuscript has been revised well. I think this manuscript is acceptable.